# Family-wide analysis of human macrodomains reveals novel activities and identifies PARG as most efficient ADPr-RNA hydrolase
Lisa Weixler [1,3,4], Roko Žaja [1,4,5] ✉, Nonso J. Ikenga[1], Jonas Siefert[1], Ganga Mohan[1], Gülcan Aydin[1],
Sven Wijngaarden[2], Dmitri V. Filippov[2], Bernhard Lüscher[1] & Karla L. H. Feijs-Žaja [1,5] ✉

ADP-ribosylation is well-known as protein posttranslational modification and was recently also identified as RNA posttranscriptional modification. When macrodomain proteins were identified as protein ADP-ribosylhydrolases, several ADP-ribosylation substrates were not yet identified. Therefore, the majority of macrodomain-containing proteins have not been tested towards these additional substrates and were considered to be inactive. Here, we compare in vitro activities of the human macrodomains on a range of ADP-ribosylated substrates. We confirm recent findings that PARP9macro1 and PARP14macro1 can remove ADP-ribose from acidic residues and provide evidence that also PARP14macro2 and PARP15macro2 can function as ADP-ribosylhydrolases. In addition, we find that both PARP9macro1 and PARP14macro1 are active as ADPr-RNA decapping protein domains. Notwithstanding these in vitro activities, our data furthermore indicate that in HEK293 cells, PARG is the major ADPr-RNA decapping enzyme. Our findings thus expand the spectrum of known catalytic activities of human macrodomains and demonstrate their different efficiencies towards nucleic acid substrates.

ADP-ribosylation has been studied for decades as posttranslational protein modification[1]. ADP-ribosyltransferases of the ARTD family, also known as PARPs, can modify target proteins either with poly(ADP-ribose) (PAR) or with mono(ADP-ribose) (MAR) in a transfer reaction that consumes NAD⁺ (Ref. 2). ADP-ribosylation of proteins appears to be involved in a plethora of processes, ranging from DNA damage repair to immune response and metabolism[1]. First described as a modification of glutamate and arginine, later work showed that other amino acids can also be ADP-ribosylated, including cysteine, serine, histidine, and tyrosine[3–6]. Arginine ADP-ribosylation is mediated by enzymes of the ARTC family, which reside on the cell membrane and modify extracellular targets[7]. Glutamate ADP-ribosylation is mediated by several enzymes of the ARTD family, including PARP10 and PARP1[3,8]. PARP1 normally poly(ADP-ribosyl)ates (PARylates) its targets, but when paired with its cofactor HPF1, its specificity is redirected towards serines, which it MARylates[4]. Mass spectrometry-based studies have identified thousands of proteins that become ADP-ribosylated

on serine residues in response to DNA damage[6,9,10]. Most ARTs have either not been investigated in detail yet, or appear to be able to modify diverse amino acids[11]. An example is PARP7, which was reported to modify mainly cysteine residues and, to a lesser extent, arginine and tyrosine[5] as well as glutamate and aspartate[12]. For most of these ADP-ribosylation sites, the consequences for the modified proteins are not yet understood. Recently, also nucleic acids were identified as substrates of specific PARPs, thereby further expanding the range of action of the ARTD family[13–16].

ADP-ribosylation of proteins and nucleic acids is reversed by enzymes belonging to two structurally distinct protein families, the macrodomain-containing proteins (MDCPs) and the ADP-ribosylhydrolases (ARHs)[17–19]. Thus far, investigations of the catalytic activities of the hydrolases have focused on the macrodomain-containing hydrolase PARG, which is known best as degrader of PAR[20], and ARH3, which reverses both PARylation as well as MARylation of serine residues[9,21]. MACROD1, MACROD2, and TARG1 were identified as hydrolases of MARylated acidic residues[22–24],

[1]Institute of Biochemistry and Molecular Biology, Pauwelsstraße 30, RWTH Aachen University, Aachen, Germany. [2]Leiden Institute of Chemistry, Leiden University Department of Bioorganic Synthesis, Einsteinweg 55, Leiden, The Netherlands. [3]Present address: Institute for Clinical Chemistry and Clinical Pharmacology, Venusberg-Campus 1, University Hospital Bonn, Bonn, Germany. [4]These authors contributed equally: Lisa Weixler, Roko Žaja. [5]These authors jointly supervised this work: Roko Žaja, Karla L. H. Feijs-Žaja. ✉e-mail: rzaja@ukaachen.de; kfeijs@ukaachen.de

whereas ARH1 reverses MARylation of arginine[25]. For ARH2, no activity has been reported so far. Although the macrodomain fold is highly conserved, it is not always apparent from the sequence alone[20]. Therefore, additional MDCPs may be identified based on structural analysis in the future.

Reversal of ADP-ribosylation appears to be essential for cell viability. MACROD1, MACROD2, and TARG1 are expressed in different tissues and localized to various compartments, implying that despite their similar catalytic activities, they have unique roles to fulfill within the cell[26]. Several studies have linked MACROD1 and MACROD2 to cancer[27], whereas a mutation in TARG1 leads to a neurodegenerative phenotype[22]. Loss of functional ARH3 leads to neurodegenerative phenotypes[28,29], while PARPG kockout is embryonically lethal[30]. Unperturbed hydrolase activity thus appears important, yet it is not well understood how the hydrolases function at a biological level, nor has their activity been systematically compared. In addition to these dedicated hydrolases, several macrodomains exist in the context of ARTs. While PARP9 and PARP15 each contain two macrodomains, PARP14 has three. Therefore, PARP9, PARP14, and PARP15 are also referred to as "macroARTs" or "macroPARPs"[31]. Recent work revealed that macro1 from PARP14 has catalytic activity in cells[32], which was already predicted a decade ago[23], as well as macro1 from PARP9[32], whereas the macrodomains from PARP15 did not show any activity. Two additional publications confirmed this activity in vitro and presented contrasting data on their activity on an ADP-ribosylated arginine substrate[33,34]. As not all macrodomains have been tested on all possible ADP-ribosylated substrates, it is possible that further macrodomains that are currently considered inactive harbour catalytic activity.

We have first performed a structural analysis of the MDCP family to identify potential new members. Next, we generated recombinant macrodomain-containing proteins and ARHs to systematically test their activity towards different protein and nucleic acid substrates. We could confirm that PARP14macro1 and PARP9macro1 show activity towards automodified PARP10 in vitro, and we could also identify PARP14macro2 and PARP15macro2 as efficient hydrolases of automodified pertussis toxin. Our investigations show furthermore that although several MDCPs harbour some catalytic activity towards ADPr-RNA, not all are equally efficient. In vitro, the main hydrolase of ADPr-RNA is PARG, followed by MACROD2 and PARP9macro1. The other hydrolases require much higher protein amounts for efficient reversal. Finally, we show that also in cells, PARG is a crucial enzyme for reversing the modification of RNA depending on the cell line used.

## Results

### Structural alignment identifies additional members of the macrodomain family

We utilized the FoldSeek tool to screen the human proteome for potential macrodomain-like folds[35]. FoldSeek facilitates rapid and accurate structural alignment of proteins. Using this tool, we identified leucine aminopeptidase 3 (LAP3) (P28838) as a protein that contains a macrodomain-like fold (Fig. 1). LAP3 is a metallopeptidase, which proteolytically processes N-termini of proteins, and which may have roles in diverse processes including immunity and cellular redox status[36]. Phylogenetic analysis, based on structural alignment with DALI, revealed that the LAP3 macrodomain clusters closely with other known MDCPs, including hydrolases such as MACROD1, MACROD2, TARG1, and PARG, the macrodomains of PARP9, PARP14, and PARP15, as well as the macrodomains associated with histones (Fig. 1a). A FoldSeek search using the predicted LAP3 macrodomain resulted in high probability hits (probability >0.80) for other macrodomains, with PARP14macro1 being the best hit (Fig. 1b). This further indicates that the predicted LAP3 domain folds as a macrodomain. As the structure of LAP3 has not yet been resolved, we utilized the predicted structure from AlphaFold (Fig. 1c) and structurally aligned the LAP3 amino acid residues 33–177 with PARP14macro1 (3q6z) (Fig. 1d, e). This structural alignment revealed that LAP3 contains a macrodomain fold similar to PARP14macro1, with root mean square deviation (RMSD) of 2.6 Å and

template modelling (TM) score of 0.71, that measures the accuracy of the global structure of the protein. However, the critical amino acid residues involved in the catalysis of ADPr removal are not conserved in LAP3 (Fig. 1d, e). Despite the probable lack of activity, we tested LAP3 in our subsequent analyses of hydrolases, where we compared the activity of the human macrodomains, including those previously thought to be inactive, such as the macrodomains from the macroARTs.

### Macrodomain activities on different ADP-ribosylated protein substrates

To be able to compare the in vitro activities of diverse macrodomains, we purified recombinant proteins from *E.coli* for the majority of human MDCPs as well as ARHs (Fig. 2a). PARP15macro1 and GDAP2 were not included in our analysis due to purification difficulties. Also in previous work where the single macrodomains from the macroARTs PARP9, PARP14, and PARP15 were co-expressed with PARP10 in cells, PARP15-macro1 could not be tested due to its instability[32]. To test hydrolase activity on a broad spectrum of substrates, we generated several ADP-ribosyltransferases with different substrate specificities. PARP10 was used to generate glutamate-linked ADP-ribosylation[8], ART2 to modify arginine[37], pertussis toxin to modify cysteine[38] and recombinant PARP1 was used to produce poly(ADP-ribose).

In accordance with previous studies[21,39–43], only PARG and ARH3 exhibit glycohydrolytic activity toward a PARylated substrate (Fig. 2b). When incubating the automodified PARP10 with the diverse hydrolases, we confirmed as expected activity of MACROD1, MACROD2, TARG1 and to a lesser extent PARG, as we have observed before[44]. However, we also note the activity of PARP9macro1 and PARP14macro1 on this substrate (Fig. 2b). This is in line with recently reported activity for these two macrodomains in cells[32] and in vitro[33,34]. It has been reported that PARP10 may be able to modify additional amino acids, including arginine and serine[45], which might explain why no complete reversal is obtained in this experiment. On the contrary, a complete reversal of arginine-ADPr is achieved by ARH1, which remains the only known intracellular enzyme capable of hydrolysing ADPr-arginine (Fig. 2b). To generate a substrate exclusively modified on cysteine, we utilised pertussis toxin (PtX), a toxin from *Bordetella pertussis* that modifies small G-protein Gi-alpha on cysteine. The PtX we used here is a truncated protein which can automodify[38]. As expected, neither the previously identified macrodomain-containing hydrolases nor the ADP-ribosylhydrolase ARH1 were active on this modification. However, we observed complete reversal of the modification by PARP14macro2 and PARP15macro2, and surprisingly also by ARH3 (Fig. 2b). This indicated that PARP9macro1 and PARP14macro1, as well as PARP14macro2 and PARP15macro2 are not merely readers of ADP-ribosylation, but actively counteract the modification.

### PARP14macro2 and PARP15macro2 are active ADP-ribosylhydrolases

To confirm that the modification we generated with PtX is cysteine-ADPr, we performed an automodification reaction and incubated it with diverse chemicals. Neutral hydroxylamine will lead to rapid loss of glutamate-ADPr and a slow release of arginine-ADPr, whereas cysteine-ADPr is sensitive to mercury chloride[46,47]. Mercury chloride caused some reduction of the signal, but not very efficiently, indicating that perhaps the automodified PtX is modified on a residue other than cysteine (Fig. 3a). Previously, a macrodomain from *Streptococcus pyogenes*, termed Spy-MacroD, was reported to have activity towards an ADP-ribosyl-cysteinyl glycosidic bond[48]. SAV0325, the *Staphylococcus aureus* homologue of SpyMacroD, contains a $Zn^{2+}$-binding motif in the proximity of the predicted ADPr-binding pocket, suggesting a diverged mechanism of hydrolysis compared to other macrodomains[49]. It has been shown that the conserved residues Cys113, His118, and Cys120 mediate $Zn^{2+}$ coordination in SAV0325[49]. We purified wild type SpyMacroD and a Cys112Ala mutant, that corresponds to Cys113 of SAV0325. To measure SpyMacroD activity, we used a cellular PtX substrate known to be modified on cysteine,

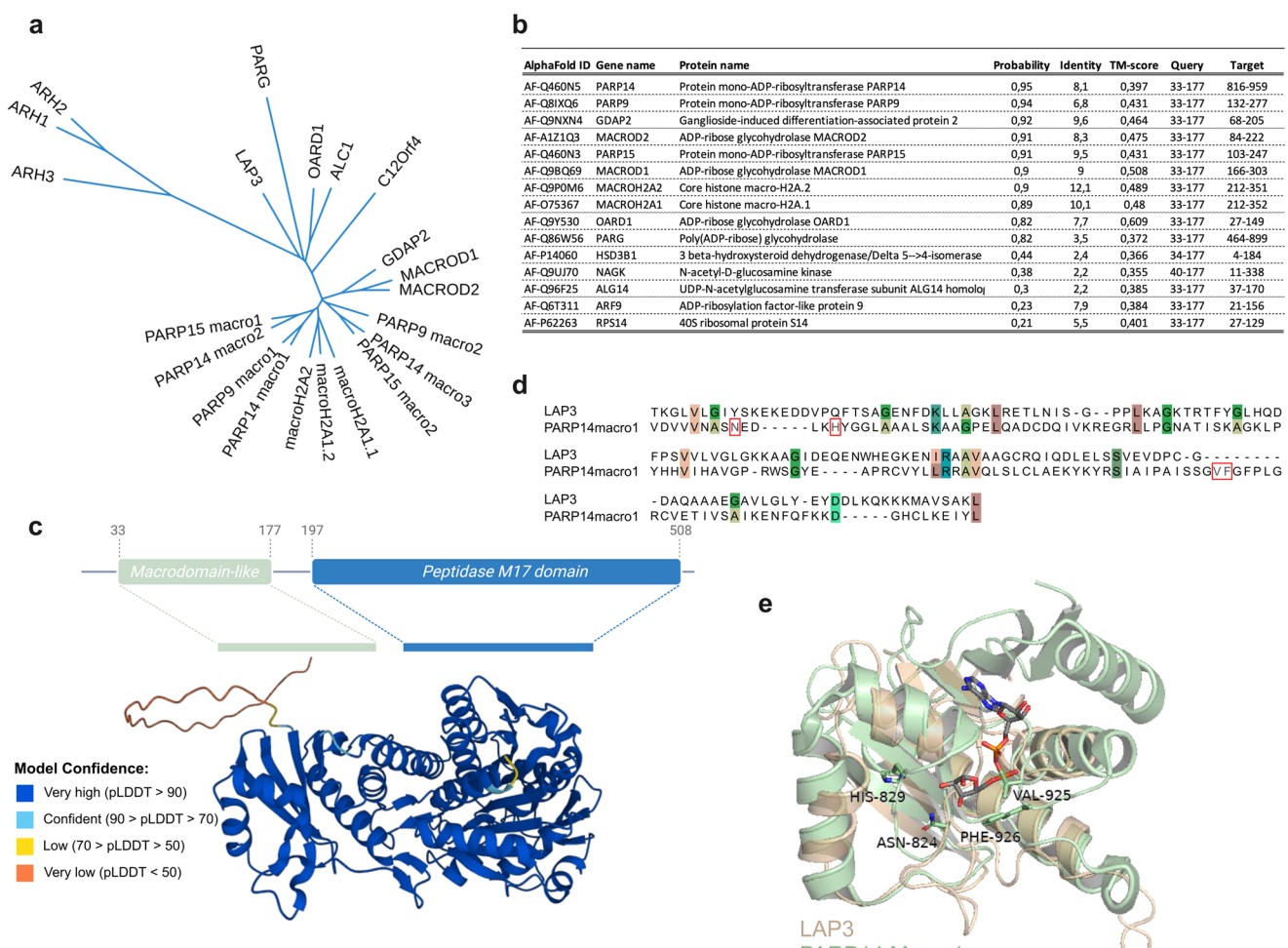

**Fig. 1 | Structural analysis of the two ADP-ribosylhydrolase families reveals an additional putative macrodomain-containing protein LAP3. a** Phylogenetic analysis of the human macrodomain proteins. **b** FoldSeek search using the LAP3 structure predicted by AlphaFold revealed several macrodomain-containing proteins as proteins most similar to the LAP3 structure. **c** Domain architecture of LAP3 with the N-terminal macrodomain-like domain ranging from amino acids 33 to 177 and the C-terminal peptidase domain ranging from amino acid 197 to 508

(top). Alphafold structure of LAP3 showing the macrodomain as well as peptidase domain of LAP3, where the colours indicate confidence of structure prediction (bottom). **d** Multiple sequence alignment of human PARP14macro1 and the proposed macrodomain of LAP3 showing conservation of proposed catalytic residues and residues involved in ADP-ribose coordination (red-framed). **e** Overlay of LAP3 and PARP14macro1 structures, with ADP-ribose modelled into the pocket.

a guanine nucleotide-binding protein G(i) subunit alpha-1 (Giα). In accordance with previous reports, Giα was modified by PtX. SpyMacroD but not the Cys112Ala mutant was able to remove this modification (Fig. 3b). We also tested whether PARP14macro2 can reverse Giα ADP-ribosylation but could not observe any activity on this substrate, which partially lost the tag used for purification, leading to two bands (Fig. 3b). Sequence alignment based on structural alignment showed that the essential Cys112 of SpyMacroD corresponds to Ser1042 in PARP14macro2 (Fig. 3c). To test the activity of SpyMacroD wildtype and mutant towards our truncated PtX, we incubated them and PARP14macro2 wildtype or mutants with automodified PtX. As observed before, PARP14macro2 efficiently reversed PtX automodification, while no activity was observed for SpyMacroD. Mutation of Ser1042 drastically reduces hydrolase activity of PARP14macro2 (Fig. 3d). In addition, mutation of Ser1047 which corresponds to Asp102 of PARP14macro2, which has been suggested to be important to cytalyse *O*-acetyl-ADP-ribose deacetylation[50], also resulted in decreased activity of PARP14macro2 toward automodified PtX (Fig. 3d). These data confirm that the intact ADPr binding pocket is needed for PARP14macro2 activity. To exclude that hydrolytic activities of PARP14macro2 and PARP15macro2 toward automodified PtX are an artifact caused by high enzyme concentrations, we titrated the amount of PARP14macro2 and PARP15macro2 from

0.1 μM to 4 μM protein (Fig. 3e). Both macrodomains remove the modification almost entirely within 30 min at a concentration of 0.25 μM. These experiments show that that the pertussis toxin automodification occurs on a different residue compared to substrate modification, which is not glutamate, arginine or cysteine.

ARH3 is highly active on Ser-ADPr[9,51], raising the possibility that perhaps the automodified PtX is modified on a serine residue. To explore the specificity of PARP14macro2 and PARP15macro2, we tested their activity, as well as that of ARH3 and SpyMacroD, on two synthetic ADPr-oligopeptides containing Cys-ADPr and Ser-ADPr[48] (Fig. 3f). To measure activity of hydrolases, ADPr-peptides were first incubated with hydrolase and then released ADPr was converted to ribose-5-phosphate and AMP by the NUDT5 enzyme. AMP was then quantified using an AMP-Glo assay[52]. As previously reported, ARH3 is highly active toward Ser-ADPr, whereas SpyMacroD efficiently removes ADPr from Cys, confirming earlier findings[48], but not from serine. PARP14macro2 and PARP15macro2 did not show activity towards either of the tested peptides. In a test with all hydrolases on a serine-, arginine-, or cystine-ADPr peptide, we see only the expected activity of ARH1 on arginine-ADPr and ARH3 on serine-ADPr (Supplementary Fig. 1). In conclusion, we could show that PARP14macro2 and PARP15macro2 possess hydrolase activity but were not able to decipher the specificity of these macrodomains.

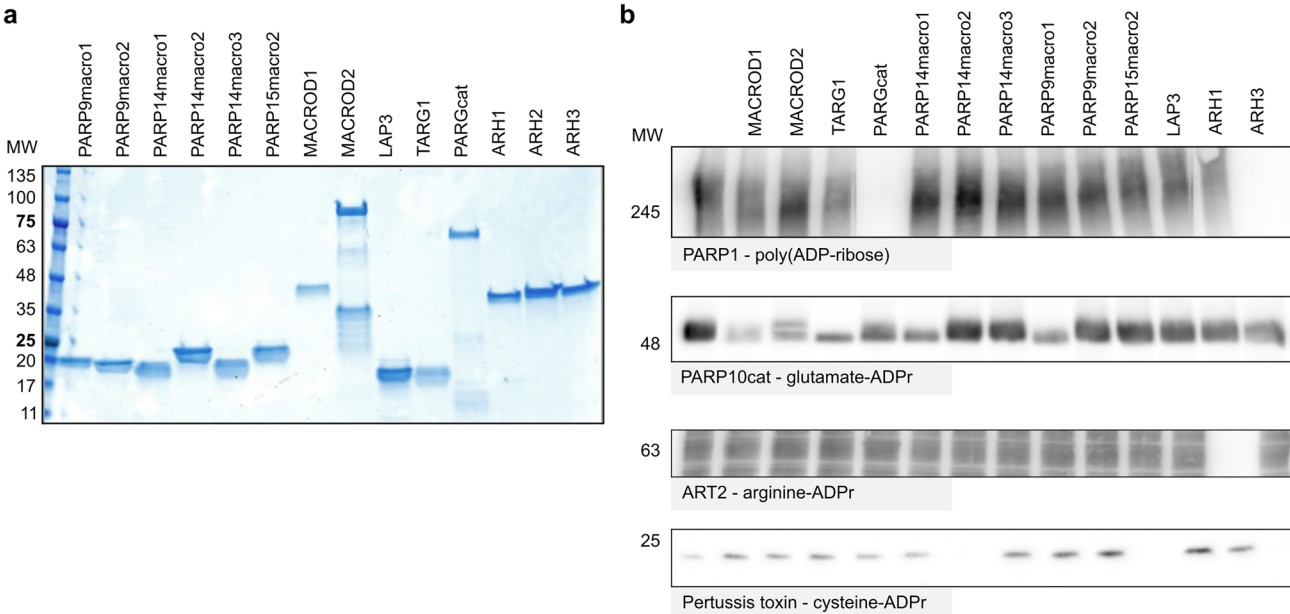

**Fig. 2 | Specific macrodomains of PARP9, PARP14 and PARP15 are active enzymes. a** Known and putative ADP-ribosylhydrolases were expressed as His/GST fusion protein in *E. coli* and purified. Purified proteins were separated using SDS gel electrophoresis and visualized using Coomassie blue. **b** ADP-ribosylated substrates were generated using in vitro reactions with recombinant PARP1, PARP10, ART2, or pertussis toxin in the presence of NAD[+]. For ART2, a cytosolic HeLa cell extract was added to provide substrates for the ADP-ribosylation reaction, whereas PARP10, PARP1, and pertussis toxin automodification were analysed. Following the ADP-ribosylation reactions, 1 µM of the respective hydrolases were added and incubated at 37 °C for an hour to allow reversal of the modification. Reactions were stopped by adding an SDS-sample buffer and analysed using western blot. Complete blots are shown in the supplementary material.

## Macrodomain activities on different ADP-ribosylated nucleic acid substrates

Recent research has identified nucleic acids as PARP substrates[53]. Given this discovery, we analysed the activity of our panel of putative hydrolases on 5'-ADP-ribosylated single-stranded RNA (Fig. 4a). Previous studies have demonstrated that MACROD1, MACROD2, TARG, and PARG remove ADP-ribose from all examined nucleic acid substrates[14,15,53] and more recent work has revealed that macro1 from PARP14, previously thought to be inactive[54], functions as hydrolase of ADPr-RNA[33,34]. Our findings align with this, confirming that PARP14macro1 is active on ADP-ribosylated ssRNA. In addition, we identified PARP9macro1 as a novel ADPr-RNA decapping enzyme. On the contrary, PARP15macro2 as well as LAP3 displayed no activity on any of the tested ADP-ribosylated RNA substrates. In a recent study analysing PARP14macro1 activity on ADPr-RNA, 4 µM of macrodomain was utilised in in vitro hydrolase assays[33]. Although enzymes are effective at picomolar or nanomolar concentrations in vivo, in vitro studies often necessitate higher enzyme concentrations due to the lack of biological complexity. Nonetheless, at 1 µM, we noted that PARP9macro1 is more active than PARP14macro1. To enable a more detailed analysis and to approach a more physiological range of enzymatic reactions, we titrated protein concentrations from 100 nM to 2 µM and assessed ADPr loss as a function of input protein quantity during the 30 min reaction (Fig. 4b, c). In addition to PARP9 and PARP14 macrodomains, we also tested the previously characterized macrodomain-containing hydrolases, MACROD1, MACROD2, TARG1 and PARG to be able to compare their efficacies. At a concentration of 100 nM, PARP9macro1 removed approximately 80% of ADP-ribose, whereas no ADP-ribose removal was observed with PARP14macro1 at the same concentration. Similar reductions were observed with five times higher concentrations of MACROD1 and TARG1, while 2 µM of PARP14macro1 was required to achieve 70% ADP-ribose removal. MACROD2 and PARGcat efficiently hydrolysed ADPr-RNA at 100 nM, the lowest concentration tested. These results indicate that the tested macrodomains exhibit varying activity levels towards ADPr-RNA (Fig. 4b), prompting further exploration into the differences between these macrodomain-containing hydrolases. To illustrate these differences, we graphically displayed de-capping activity by quantifying ADPr-RNA turnover (Fig. 4c). MACROD2 and PARG are not included in this analysis, as the substrate was almost fully decapped at the lowest concentration tested.

## PARG is the most effective RNA ADP-ribosylhydrolase in vitro

We next characterised PARG and MACROD2 in more detail since these hydrolases show the highest efficiency of ADPr-RNA hydrolysis at the concentrations used in our initial ADPr-RNA hydrolase test (Fig. 4). We first titrated the amount of enzyme used in the assay. It becomes clear that MACROD2 is more efficient than previously tested hydrolases, with 10 nM enzyme still removing 60% of ADP-ribose in 30 min of incubation (Fig. 5a, b). However, PARGcat removed 60% of ADPr at a 1 nM enzyme concentration in the same incubation period, indicating even higher processivity (Fig. 5a, b). To compare the activities of PARG and MACROD2 in more detail, we monitored the de-capping of ADPr-RNA over time at a fixed saturating substrate concentration of 1 µM of RNA oligo (Fig. 5c, d). Based on our protein titration assay we used a 250 pM concentration of PARG and a 10 nM concentration of MACROD2. At a PARG concentration of 250 pM, the maximum velocity ($V_{max}$) was observed to be 0.7 µM/min. Since the Michaelis constant ($K_m$) cannot be determined from time dependent experiments, we determined the apparent rate constant ($K_{app}$), which reflects $K_m$ under the given reaction conditions. This allows for a direct comparison of PARGcat and MACROD2 activities. $K_{app}$ for PARG was approximately 5.44 µM while the calculated turnover number ($k_{cat}$) was 46.7 s$^{-1}$ (Table 1). The catalytic efficiency ($k_{cat}/K_{app}$), was $8.59 \times 10^6$ M$^{-1}$s$^{-1}$, reflecting the very high efficiency of PARG in decapping ADPr-RNA. In contrast, the MACROD2 exhibited a $V_{max}$ of 0.749 µM/min and $K_{app}$ of 4.70 µM, which is within a similar range as observed for PARG. However, MACROD2 had a $k_{cat}$ of 1.25 s$^{-1}$ and its catalytic efficiency was calculated to be 265.7 M$^{-1}$s$^{-1}$, nearly two orders of magnitude lower than that of PARG. The substantially higher catalytic efficiency of PARG compared to MACROD2 suggests that PARG is better suited for the efficient catalysis of ADPr-RNA under the tested conditions, making it the most active ADPr-RNA hydrolase in vitro. These calculations are based on slot blots, which are

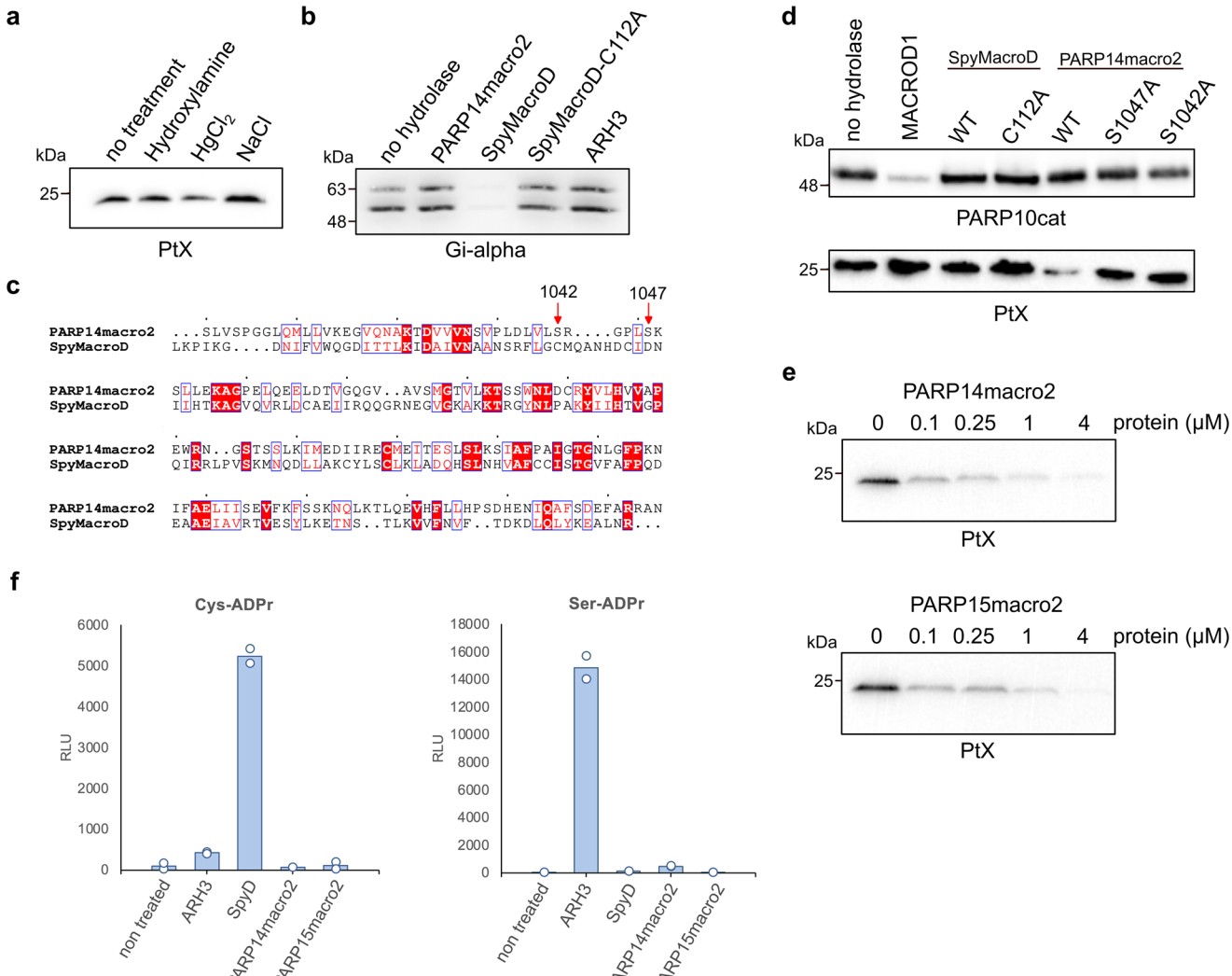

**Fig. 3 | PARP14macro2 and PARP15macro2 are efficient hydrolases of auto-modified pertussis toxin. a** Automodified pertussis toxin was incubated with neutral hydroxylamine, mercury chloride, or sodium chloride for 3 h at 37 °C. Susceptibility to chemical treatment was analysed using SDS-PAGE followed by western blotting with an ADPr antibody. **b** Truncated pertussis toxin was incubated with Gi-alpha for 30 min in the presence of NAD⁺, followed by incubation with indicated putative hydrolases. **c** Sequence alignment shows the conservation of catalytic residues and the residues involved in ADP-ribose coordination of human PARP14macro2 compared to *Streptococcus pyogenes* macro (SpyMacroD). The numbered residues are serine 1042 and serine 1047 of PARP14, which correspond to cysteine 112 and asparate 121 of SpyMacroD. **d** PARP10cat (top) or pertussis toxin (bottom) were automodified and incubated with MACROD1, SpyMacroD2 wild-type and mutant or PARP14macro2 wildtype and mutants. Samples were analysed on western blots. **e** Automodified mutant pertussis toxin was incubated with indicated amounts of PARP14macro2 and PARP15macro2 for 30 min at 37 °C. Reversal of modification was analysed using SDS-PAGE and western blotting. **f** An AMP-Glo assay was performed using specific ADPr-peptides, which were incubated with the indicated hydrolases and NUDT5.

only semiquantitative and can thus only be used to compare the activities of the enzymes tested here.

## PARG catalytic mechanism towards ADPr-RNA differs from its PAR hydrolysis activity

PARG's activity towards PAR has been well studied, with two key glutamates in the catalytic centre contributing to PAR hydrolysis (Fig. 6a)[20]. To test whether these residues are also essential for ADPr-RNA hydrolysis, we generated PARGcatE755A, PARGcatE756A as well as a double glutamate mutant and first tested their activity on PAR. We could confirm that mutating either of these glutamates results in a loss of glycohydrolytic activity on PARylated PARP1 (Fig. 6b). However, when applied at the same concentration, the mutant PARG proteins remained active on ADPr-RNA, hinting that the catalytic mechanism for RNA and protein substrates might differ (Fig. 6c). In a protein titration experiment, it became apparent that the E755A mutant lost some of its activity towards ADPr-RNA, whereas E756A did not (Fig. 6d). To directly compare their activities, we monitored the de-

capping of ADPr-RNA over time at a fixed, saturating substrate concentration of 1 μM RNA oligo in the presence of 250 nM PARGcat-E755A mutant (Fig. 6e). Unlike wild type PARGcat, the E755A mutant exhibited altered kinetic behaviour, with a similar $V_{max}$ of 1.1 μM/min but a substantially increased $K_{app}$ of 28.9 μM, indicating diminished substrate affinity. The $k_{cat}$ was notably reduced to 0.071 s⁻¹, resulting in a drastic decrease in catalytic efficiency to 2452 M⁻¹s⁻¹, an order of magnitude compared to wild type PARGcat. Interestingly, the residual activity of the E755A mutant on ADPr-RNA, despite its significant kinetic impairment, contrasts sharply with its inability to catalyze PAR degradation even at higher enzyme concentrations (up to 4 μM). This observation suggests a potential divergence in the catalytic mechanisms governing ADPr-RNA de-capping and PAR degradation by PARG. PAR chains are formed by *O*-glycosidic bonds between ribose units, whereas ADP-ribose on RNA is attached to a phosphate through an O-glycosidic phosphoester bond. Therefore, the bonds hydrolysed in these substrates are different (Fig. 6f). It is not surprising that other residues may be essential or that different catalytic mechanisms are

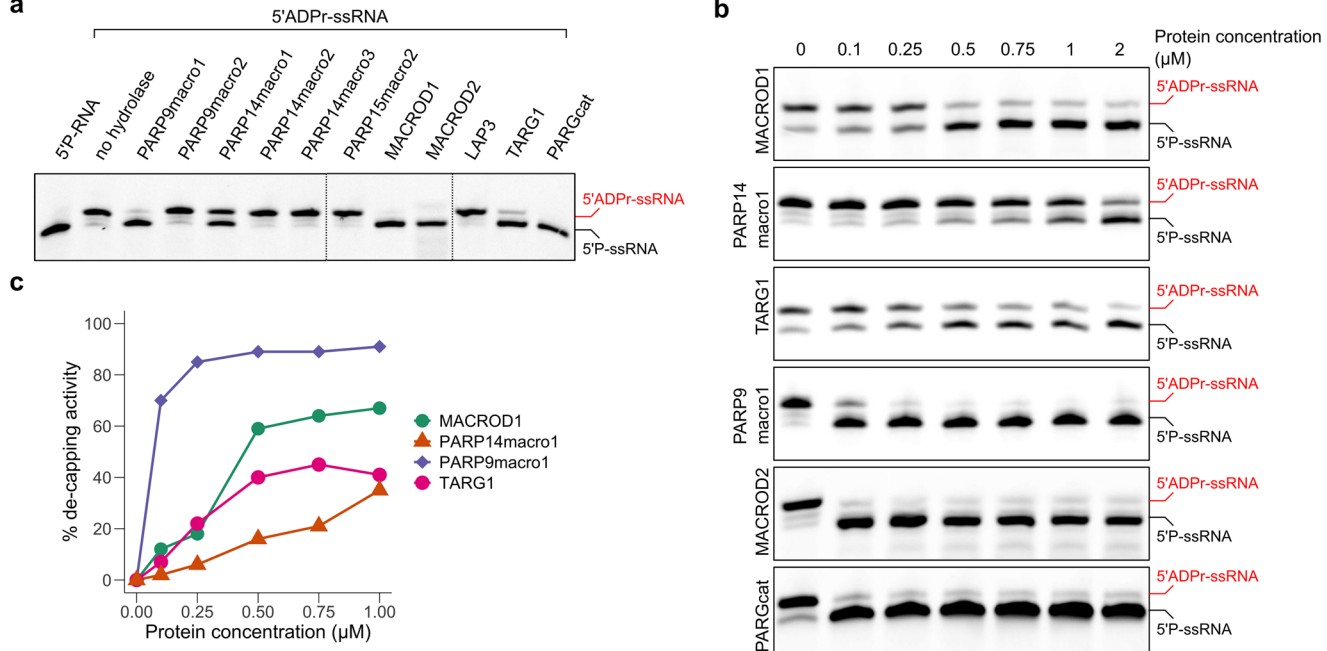

**Fig. 4 | Distinct efficiency macrodomain hydrolases in ADPr-RNA hydrolysis.**
**a** A synthetic 5′-phosphorylated ssRNA oligo (5′P-GUU UCG GAU CGA CGC-3′ Cy3) was ADP-ribosylat with TRPT1 at 37 °C for 30 min to generate ADPr-capped substrate. Proteinase K was added to avoid protein/RNA co-precipitation during the following column purification. 0.25 μM purified ADPr-RNA oligo was subsequently incubated with 1 μM of the indicated hydrolases at 37 °C for 40 min. Reactions were separated via urea-PAGE. **b** An ADPr-RNA oligo was generated as in (**a**). 1 μM oligo was incubated with increasing amounts (0/0.1/0.25/0.5/0.75/1/2 μM) of the indicated hydrolases at 37 °C for 40 min. Reactions were separated via urea-PAGE. **c** The intensity of the ADPr-RNA bands observed in (**b**) was measured and quantifed. The reversal of the modification was plotted.

employed for the hydrolysis of these diverse substrates. PARG is the only protein in the MDCP family for which inhibitors have been developed. Given that the catalytic mechanism of PAR and ADPr-RNA hydrolysis appears to differ, we next tested whether these PARG inhibitors could effectively inhibit ADPr-RNA hydrolysis in cells.

## PARG is a key RNA ADP-ribosylhydrolase in cells

Previously, to detect ADP-ribosylation of RNA in mammalian cells, we used a cocktail of siRNA to knock down ARH3, TARG1, and PARG as potential ADPr-RNA hydrolases[15]. As MACROD1 is strictly mitochondrial and MACROD2 is not ubiquitously expressed[26], we did not remove these in our previous work before analysing cellular ADPr-RNA content. In a TRPT1-overexpression context, knockdown of ARH3, TARG1 and PARG led to the largest increase in RNA modification in HeLa cells[15]. Considering the lower in vitro efficiency of ARH3 and TARG1, we now transfected HEK293 cells with siRNA against either TARG1, PARG, or ARH3 and determined the levels of RNA ADP-ribosylation by slot blotting. Although TARG1 and ARH3 knockdown led to an increase in RNA modification, PARG knockdown led to the largest increase in RNA ADP-ribosylation (Fig. 7a). We also incubated RNA purified from cells with recombinant MACROD1 to reverse the modification, confirming that the signal detected was ADP-ribose capped RNA. Although this hints at the significance of PARG as hydrolase of ADPr-RNA in HEK293 cells, the increased RNA MARylation could also be the result of secondary effects caused by prolonged PARG depletion. To address this, we treated cells with different concentrations of PARG inhibitor PDD00017273 (PARGi). The PARG inhibitor caused a concentration-dependent increase in cellular ADPr-RNA level with maximal levels at 5 μM (Fig. 7b). The observed increase in ADPr-RNA levels after PARGi treatment was also time-dependent. A significant increase in ADPr-RNA levels could be observed as early as 15 min after the addition of PARGi to the cultured cells, indicating a very fast turnover of ADPr-RNA in cells (Fig. 7b). Maximal ADPr-RNA levels were reached within 60 min. However, this effect was highly cell line dependent, as apparent from an analysis of ADPr-RNA content in diverse, routinely cultured cell lines in our

lab, in response to PARGi (Fig. 7c). In HEK293 cells, a considerable increase is visible upon PARG inhibition, whereas the detected increase in other cell lines was less pronounced. When we immunoprecipitated GFP-PARG from HEK293 cells and used this in an in vitro assay, we could significantly reduce its activity with the inhibitor (Fig. 7d). To determine whether the increase in detectable RNA modification in specific cell lines might correlate with PARG protein expression, we analysed PARG expression levels in a set of cell lysates (Fig. 7e). No clear correlation can be observed between cellular PARG levels and ADPr-RNA levels in response to PARG inhibitor, as for example HeLa and HEK293T cells express PARG at similar levels. The difference in response could possibly be explained by HEK293T cells having higher basal PARG activity, or by the differential expression of other hydrolases in specific cell types. From our analyses, it thus appears that PARG is the most efficient and possibly main hydrolase of ADPr-RNA in certain cell lines.

## Discussion

Our work reveals the catalytic activity of specific macrodomains, including some that were hithertho believed to be inactive. The newly identified LAP3 does not show activity towards any of the substrates tested and may therefore either not function as a hydrolase or be active on an as yet unknown substrate. We could confirm that PARP9macro1 and PARP14-macro1 are active on automodified PARP10. In an earlier experiment using murine PARP14macro1, no such activity was seen, hinting that there may be significant differences between murine and human PARP14macro1[24]. In contrast to the recently reported activity of PARP14macro1 towards arginine-ADPr[34], we did not observe reversal of arginine-ADPr by any hydrolase other than ARH1 (Fig. 2). A second recent publication also did not measure activity towards arginine-ADPr, but only detected activity towards glutamates, consistent with our findings[33]. The hydrolases may have different activities depending on the exact substrate, which was different in these studies. In summary, we have expanded the set of mammalian ADP-ribosylhydrolases to contain two enzymes which reverse poly(ADP-ribose), five capable of reversing glutamate-ADPr, one for

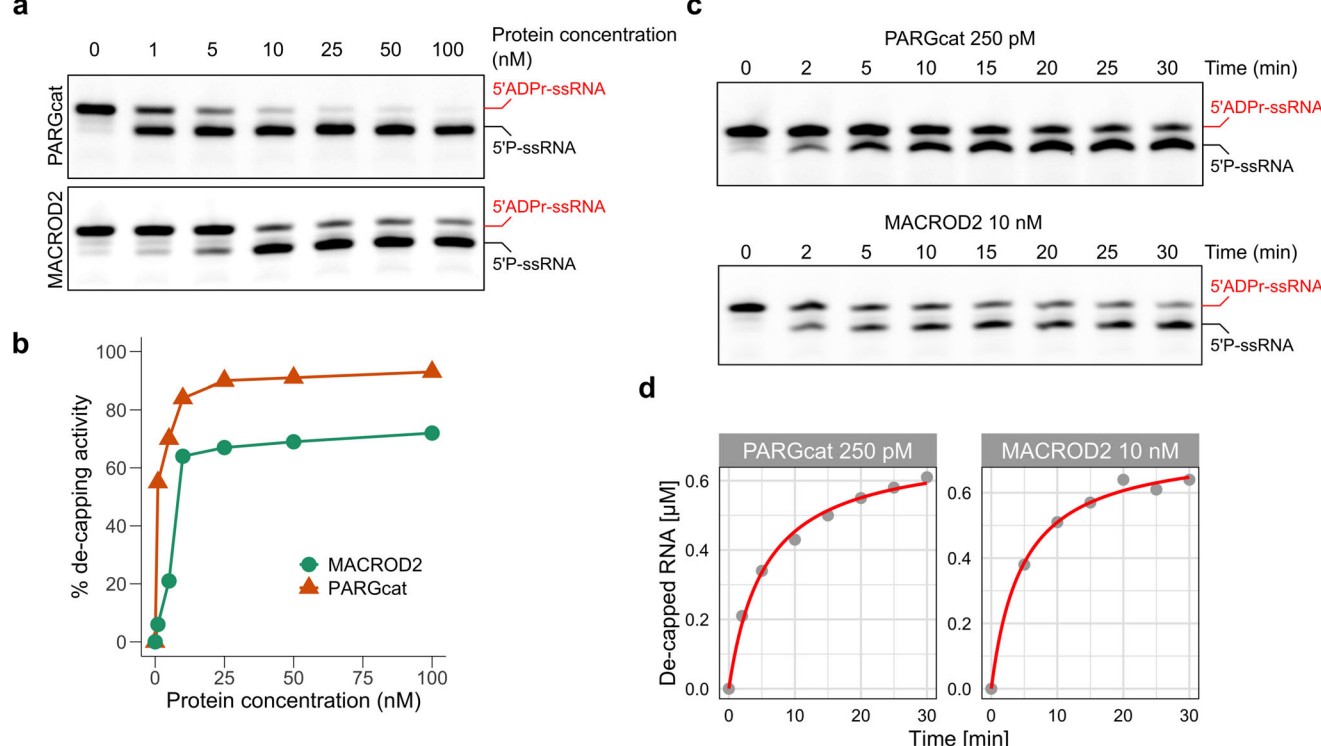

**Fig. 5 | PARG is highly active toward ADPr-RNA. a** A synthetic 5′-phosphorylated ssRNA oligo (5′P-GUU UCG GAU CGA CGC-3′Cy3) was ADP-ribosylated with TRPT1 as described. 1 μM purified oligo was incubated with the indicated hydrolases in increasing amounts: 0/1/5/10/25/50/100 nM. Reactions were separated via urea-PAGE. **b** Reversal of the modification observed in (**a**) was quantified and plotted. **c** A synthetic 5′-phosphorylated ssRNA oligo (5′P-GUU UCG GAU CGA CGC-3Cy3) was ADP-ribosylated with TRPT1. 0.5 μM modified oligo was incubated with PARGcat (250 pM) or MACROD2 (10 nM) for the indicated periods. The reaction was stopped by proteinase K treatment and subsequent heating in RNA loading dye. Reactions were separated via urea-PAGE. **d** Reversal of the modification observed in (**c**) was quantified and plotted.

**Table 1 | Kinetic parameters ADPr reversal**

|  | $k_{app}$ | $k_{cat}$ | $k_{cat}/k_{app}$ | $V_{max}$ |
|---|---|---|---|---|
| MACROD2 | 4.70 μM | 1.25 s$^{-1}$ | 265.7 M$^{-1}$s$^{-1}$ | 0.749 μM/min |
| PARGcat | 5.44 μM | 46.7 s$^{-1}$ | 8.59×10$^6$ M$^{-1}$s$^{-1}$ | 0.7 μM/min |

serine-ADPr, one for arginine-ADPr, and the two novel macrodomains reversing another, as yet unknown, modification (Table 2). For PARP14-macro2 and PARP15macro2, further biochemical characterisation is required to uncover the nature of substrates potentially hydrolysed. It will also be essential to assess the relevance of these novel activities in the full-length protein context and in cells. Mainly for ARH3, studies have been performed to determine its contribution to the serine-ADPr landscape, which have shown its essential role in the homeostasis of this modification[6,9,55]. Such mass spectrometry-based studies, in which individual hydrolases are removed from cells, will be required to determine the contribution of PARP14macro2 and PARP15macro2 domains as active hydrolases in cells. In PARP14 and PARP15, ADP-ribosyltransferase and hydrolase activities are combined. This combination of activities represents a challenge, as hydrolase and transferase activities must be separately regulated to study either in their cellular context. It is, in theory, possible to express single domains and analyse their activity in cells. However, it is doubtful that without the context of their native protein, they obtain the same localisation and interactome as the native protein. Ideally, future research will develop inhibitors specific for individual ARTs, as for example for PARP7 and PARP14 already exist[56,57], and also for the individual macrodomains similar to the available PARG inihibitor. By explicitly inhibiting certain functions of the proteins, the individual domains can be investigated in the context of the full-length protein. Alternatively, loss-of-function

mutations can be introduced in either macro or ART domains to analyse the function of full-length protein with loss of activity in one of its domains.

Going beyond protein ADP-ribosylation, we observed that in addition to the known hydrolases MACROD1, MACROD2, TARG and PARG also PARP9macro1 and PARP14macro1 are active on ADPr-RNA (Fig. 4). We furthermore observed that PARG is much more efficient at reversal of ADPr-RNA than any of the other enzymes tested in vitro. PARP9macro1 and MACROD2 also show increased activity compared to the other enzymes, but not to the same extent as PARG (Fig. 5). The robust activity of PARGcat towards ADPr-RNA, characterized by a high turnover number ($k_{cat}$ of approximately 46.7 s$^{-1}$) and exceptional catalytic efficiency (approximately $8.59 \times 10^6$ M$^{-1}$s$^{-1}$), underscores its efficiency in cleaving the phosphoester bond between ADP-ribose and RNA. The involvement of E755 and E756 in facilitating this process is complex, as illustrated by the differential impact of the E755A and E756A mutations. While both mutations render PARG inactive towards substrates linked by an O-glycosidic bond of PAR, their effects on ADPr-RNA processing diverge significantly. The mutation of E755A previously proposed as a key residue in ADPr binding during PAR hydrolysis by PARGcat, results in markedly diminished catalytic parameters ($k_{cat}$ of approximately 0.071 s$^{-1}$) and catalytic efficiency (approximately 2452 M$^{-1}$s$^{-1}$) in ADPr-RNA hydrolysis. This suggest a crucial role for E755 in the optimal positioning or stabilization of ADPr-RNA for catalysis. However, mutation of the catalytic E756 residue did not affect ADPr-RNA de-capping but completely inactivated PARGcat for PAR hydrolysis as shown before. This underscores a fundamental difference in how PARGcat processes ADPr-RNA and PAR. MACROD2, inherently lacking the glutamates analogous to E755 and E756 in PARG, retains efficient catalytic activity towards ADPr-RNA. This suggests that the presence of these specific residues, while crucial for PARG's activity towards PAR, may not be as critical for ADPr-RNA processing. The absence of key

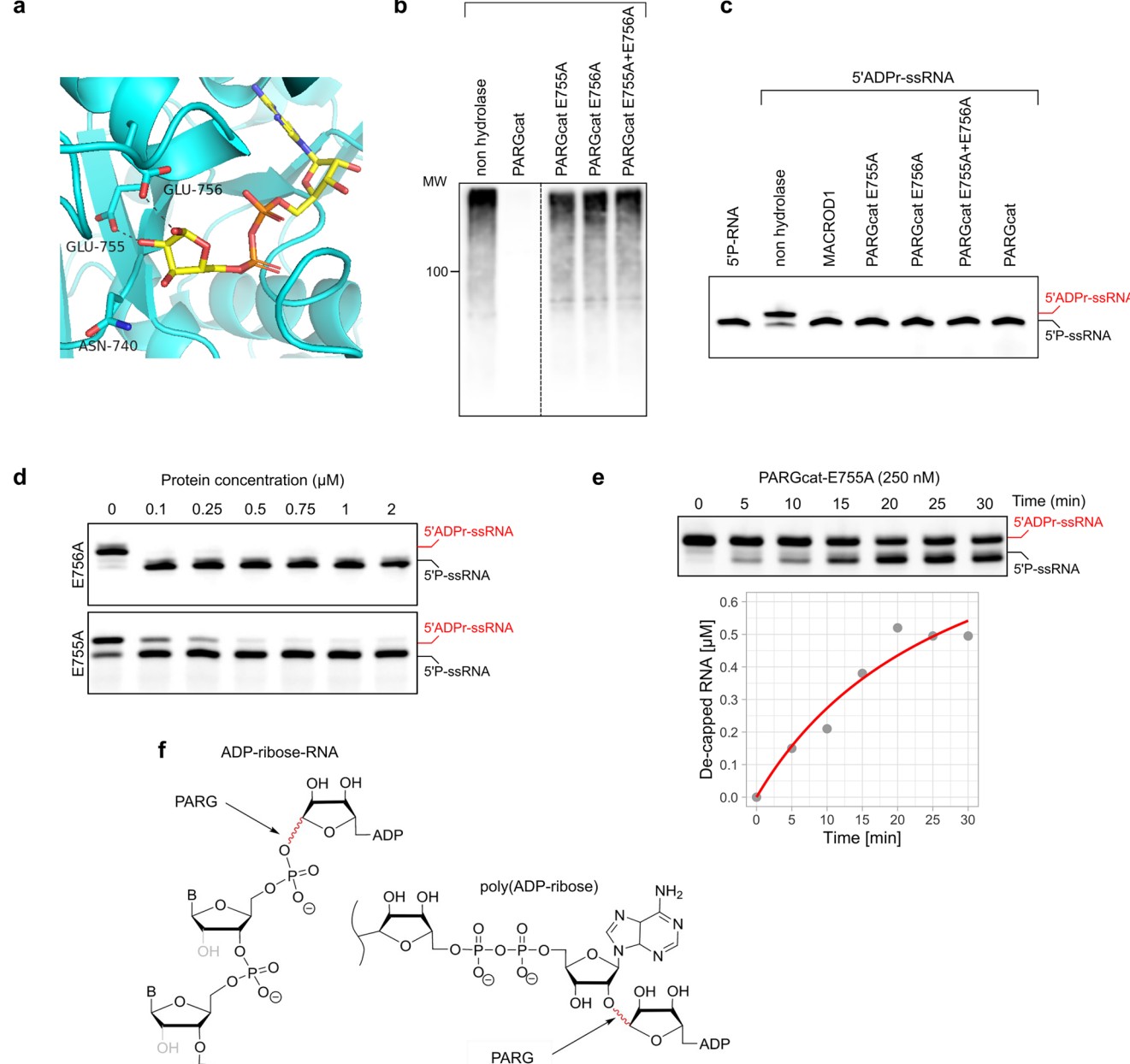

**Fig. 6 | The catalytic mechanism of PAR and ADPr-RNA reversal by PARG differ.**
**a** Structure of the PARG catalytic pocket with residues indicated that are involved in catalysis of pol(ADP-ribose). **b** PARP1 was automodified in a 30 min reaction in the presence of 500 μM NAD+. Its activity was inhibited by adding olaparib, followed by the addition of 100 nM GST-PARGcat wildtype or mutants. Hydrolase assays were performed for 30 min at 37 °C and analysed using western blotting. **c** A synthetic 5′-phosphorylated ssRNA oligo (5′P-GUU UCG GAU CGA CGC-3′Cy3) was ADP-ribosylated with TRPT1. 0.25 μM modified oligo was incubated with 2 μM of the indicated hydrolases at 37 °C for 30 min. Reactions were separated via urea-PAGE. **d** 1 μM purified oligo was incubated with indicated amounts of PARG mutants at 37 °C for 40 min. Reactions were separated via urea-PAGE. **e** 1 μM purified oligo was incubated with 250 nM PARG-E755A at 37 °C for the indicated periods. Reactions were separated via urea-PAGE, quantified and plotted. **f** Depicted are poly(ADP-ribose) and ADPr-RNA, with highlighted in red the bonds hydrolysed by PARG in those substrates.

catalytic residues in MACROD2 led to the proposal of a substrate-assisted mechanism, in which a coordinated water molecule in the active site would be catalytic[23]. The ability of MACROD2 to process ADPr-RNA efficiently hints that also the PARGcat E756A mutant does not rely on classic hydrolysis through formation of an oxocarbenium intermediate for ADPr-RNA decapping, but uses an alternative catalytic mechanism. This is possibly a substrate-assisted catalysis similar to the mechanism suggested for MACROD2. Future work using more quantitative methods to measure PARG activity can further clarify this, as well as structural studies of PARG with ADPr-RNA.

Moreover, we observed previously that MACROD2 is expressed at low levels in only a few cell lines[26]. Considering the apparent low MACROD2 expression levels and moderate activity of the other hydrolases in vitro, it is questionable how much each of these enzymes contributes to RNA demodification in cells. Removal of PARG using siRNA, as well as its inhibition with an inhibitor, led to a substantial increase in the amount of ADP-ribosylated RNA in HEK293 cells (Fig. 7). This implies that, at least under the conditions tested, PARG is the main hydrolase of ADPr-RNA in HEK293 cells. This appears in contrast to our earlier work in HeLa cells where knockdown of ARH3, TARG1 and PARG led to a more substantial

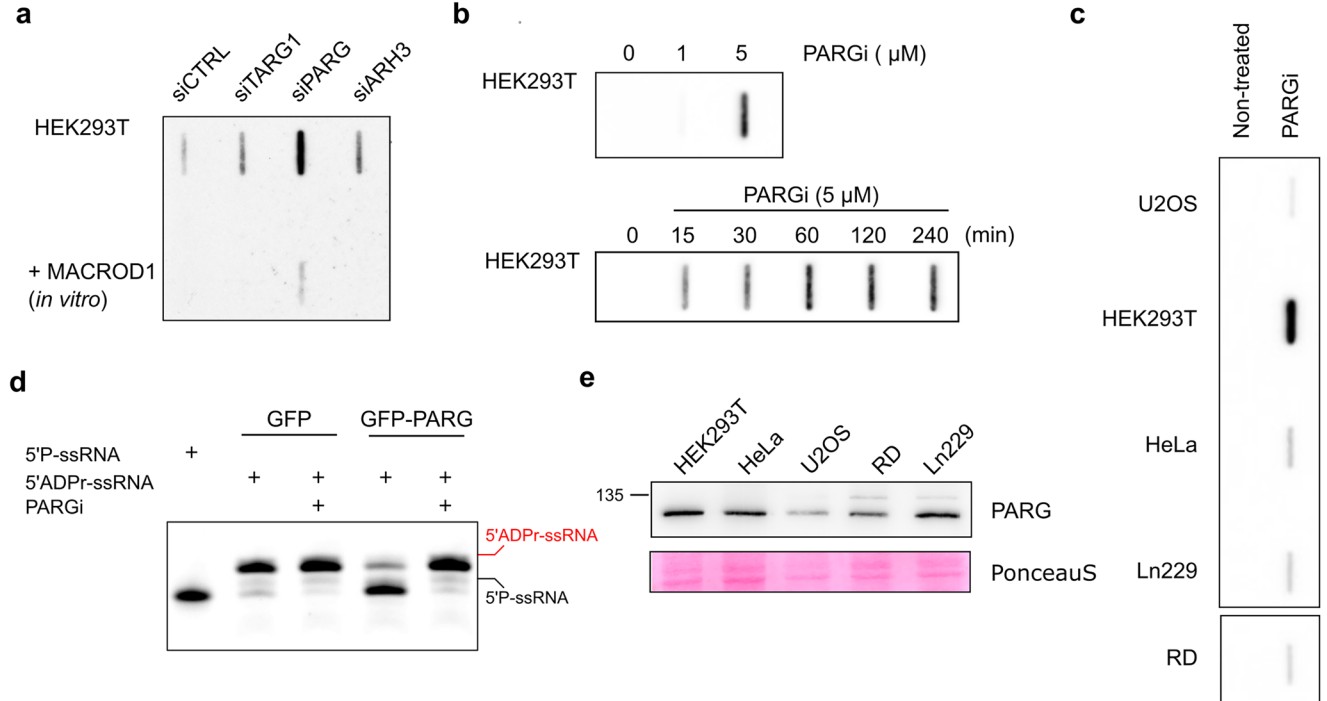

**Fig. 7 | PARG is the main ADPr-RNA hydrolase in human cells. a** HEK293T cells were reverse transfected with siRNA to knock-down TARG, ARH3 or PARG. 48 h after transfection, total RNA was isolated and incubated with MACROD1 at 37 °C for 30 min. 400 ng RNA was slot blotted and analysed with a PAR/MAR antibody. **b** HEK293 cells were incubated with 0, 1 or 5 μM PARGi for 4 h (top) or with 5 μM PARGi for the indicated periods (bottom), followed by RNA extraction and slot blotting. **c** Different cell lines treated with 5 μM PARGi for 2 h. Total RNA was isolated and aqual amounts of RNA of each sample were slot blotted. **d** GFP and GFP-PARG were transfected in HEK293T cells. After 48 h cells were lysed and the GFP-proteins enriched using a magnetic GFP-trap. GFP-PARG and GFP were incubated with 1 μM ADP-ribosylated RNA oligo (ADPr-GUU UCG GAU CGA CGC-3′Cy3) and 5 μM PARGi at 37 °C for 30 min. Reactions were separated via urea-PAGE. **e** Indicated cell lines were lysed in RIPA buffer and analysed using western blotting with a PARG antibody.

increase than knockdown of PARG alone[15]. Of note, the response to PARG inhibition is not equal for all cell lines tested. It is well possible that the isolated macrodomains tested here in vitro may have specific co-factors, which they require for full activity. The expression of these hydrolases as well as potential co-factors may vary between cell lines, and provide a tentative explanation as to why only in specific cell lines PARG inhibition suffices to upregulate RNA MARylation. Future work investigating ADP-ribosylation of RNA should consider the highly efficient reversal of this modification by PARG and possibly other hydrolases in other cell lines.

Lastly, our work provides a note of caution for future in vitro ADP-ribosylation studies: many macrodomains show activity towards ADPr-RNA when employed at high concentrations, which has led researchers to assume equal activities in the past. Having titrated their amounts, however, we could observe striking differences in efficiencies. It would be recommendable for future in vitro experiments to reduce the amounts of enzyme used to be able to observe such differences in activity. The same applies for protein deMARylation, where even after 30 min incubation with 1 μM hydrolase still residual signal is present on PARP10 (Fig. 2). Reversal of arginine-ADPr by ARH1 is complete under these conditions, as is hydrolysis of PARylation by PARG and ARH3, which implies that the reversal of PARP10 MARylation by the enzymes used is not very efficient. There are several putative explanations for this observation, such as the lack of required hydrolase cofactors, the underlying amino acid sequence being more or less suitable for hydrolase interaction, or the potential mix of MARylation sites, i.e. more than one acceptor amino acid, present on PARP10 in these conditions[11,45]. The latter was reported before to contain possibly serine and arginine in addition to glutamate, although an earlier experiment incubating automodified PARP10 with ARH1, ARH3 and MACROD1 still did not lead to full reversal of the signal[44]. Also, the potential ADP-ribosyltransferase activity on three different amino acid side chains, both regarding length and nucleophile, would need to be explained.

There are several limitations. For GDAP2 and PARP15macro1, we could not generate proteins of sufficient quality and quantity to test their activity in our in vitro assays. It therefore remains possible that also these macrodomains harbour activity towards the ADP-ribosylated substrates tested in this work. Furthermore, we have now tested the activity of isolated protein domains in in vitro reactions. In the full-length protein context their activity might be different, co-factors may influence their activity, as well as embedding in their natural environment (i.e., cellular compartment). Lastly, we tested activity on somewhat artificial substrates, such as automodified PARPs and a truncated pertussis toxin. These experiments show that, in principle, the macrodomains 2 from PARP14 and PARP15 have enzymatic activity but leaves open the exact identify of their substrate as well as relevance of this activity in cells, which future work needs to address. For PARG, we have observed that mutation of residues critical for PAR hydrolysis has less effect on its ADPr-RNA hydrolytic capacities. We have therefore suggested that PARG possibly employs a substrate-assisted catalysis mechanism to reverse RNA ADP-ribosylation. At this moment we do not have structural data to confirm this, which should be addressed in future crystallographic studies. Finally, the calculations of PARG activity render approximations of PARG activity due to the fact that slot blots are semi-quantitative and depend for example on antibody efficiency.

## Material and methods
### Plasmids and site-directed mutagenesis
Several expression constructs were provided by other researchers: Fritz Koch-Nolte shared the ART2 expression construct[58], Patricia Korn shared PARP9 and PARP15 macrodomain expression constructs[59], Lari Lehtiö provided truncated pertussis toxin, PARGcat and ARH1 expression constructs[60]. MACROD2, TARG1, ARH2 and ARH3 plasmids were generated in our lab before[26,44] as were TRPT1 constructs[15]. The LAP3 macrodomain as well as NUDT5 were cloned using a gBlock (IDT) with

**Table 2 | Overview of the macrodomain-containing protein and ADP-ribosylhydrolase families and their to-date known catalytic activities**

| | | PAR | MAR | ADPr-RNA | ADPr-DNA | Ref. |
|---|---|---|---|---|---|---|
| Macrodomain-containing | MACROD1 | – | ++ (E) | + | + | This work 14,23,44; |
| | MACROD2 | – | ++ (E) | +++ | + | This work 14,24,53; |
| | TARG1 | – | ++ (E) | + | + | This work 14,23,44,53; |
| | PARG | +++ | + | +++ | +++ | This work 14,44,53,64; |
| | GDAP2 | n.d. | n.d. | n.d. | n.d. | |
| | PARP9macro1 | – | + (E) | ++ | + | This work 32; |
| | PARP9macro2 | – | – | – | – | This work 32; |
| | PARP14macro1 | – | + (E) | + | + | This work 32; |
| | PARP14macro2 | – | ++ (?) | – | – | This work |
| | PARP14macro3 | – | – | – | – | This work 32; |
| | PARP15macro1 | n.d. | n.d. | n.d. | n.d. | |
| | PARP15macro2 | – | ++ (?) | – | – | This work 32; |
| | macroH2A1.1 | – | – | n.d. | n.d. | 23,65 |
| | macroH2A1.2 | – | – | n.d. | n.d. | 65 |
| | macroH2A2 | – | – | n.d. | n.d. | 65 |
| | LAP3 | – | – | – | – | This work |
| ADP-ribosylhydrolase | ARH1 | – | +++ (R) | – | – | This work 14,25,44,53; |
| | ARH2 | – | – | – | – | 14,53 |
| | ARH3 | +++ | +++ (S) | ++ | ++ | This work 14,21,44,53; |

*n.d.* not determined.

**Table 3 | conditions for recombinant protein expression**

| Protein | Plasmid | Bacterial strain | OD$_{600}$ at induction | IPTG [mM] | Time | Temp. | Source |
|---|---|---|---|---|---|---|---|
| MACROD1 | pNH-TrxT-MACROD1_77-325 | BL21-CodonPlus (DE3)-RIL | 0.6 | 0.4 | 16 h | 22 °C | 50 |
| MACROD2 | pDEST17-MACROD2 | BL21-CodonPlus (DE3)-RIL | 0.6 | 0.4 | 16 h | 22 °C | 24 |
| TARG1 | pDEST17-TARG1 | BL21-CodonPlus (DE3)-RIL | 0.6 | 0.4 | 16 h | 22 °C | 66 |
| PARGcat | pNH-TrxT_hPARG_cat_448-976 | Rosetta™(DE3)pLysS | 0.6 | 0.4 | 16 h | 18 °C | 60 |
| ARH1 | pNIC-Bsa4_hARH1 | BL21-CodonPlus (DE3)-RIL | 0.6 | 1 | 16 h | 18 °C | 60 |
| ARH3 | pDEST17-ARH3 | BL21-CodonPlus (DE3)-RIL | 0.6 | 1 | 6 h | 16 °C | 8 |
| PARP10_cat | pGST-PARP10_818-1025 | BL21-CodonPlus (DE3)-RIL | 0.6 | 1 | 16 h | 18 °C | 8 |
| mART2.2 | pASK60-ART2.2-FlagHis6x | Lemo21(DE3) E. coli | 0.6 | 1 | 16 h | 16 °C | 58 |
| Pertussis toxin_cat | pET15b-rPtxS1 | BL21-CodonPlus (DE3)-RIL | 0.6 | 0.4 | 16 h | 18 °C | 38 |

Gateway compatible extensions to allow insertion into pDONR221, followed by an LR reaction to transfer the inserts to pDEST17. The full-length PARG was ordered as gBlock from IDT, inserted into pDONR221 and transferred to pcDNA5/FRT/TO-N-mEGFP. To generate PARG cat mutant plasmids, we used the NEB Q5 Site-Directed Mutagenesis kit. Plasmids we generated will be available from Addgene.

**Protein purification**
The recombinant expression and purification of the ADP-ribosyltransferases PARP10cat and ART2, as well as of the ADP-ribosylhydrolases MACROD1, MACROD2, TARG1, PARGcat, ARH1, ARH2 and ARH3 was performed as described before, with the table of expression and purification conditions copied below (Tables 3 and 4)[44]. Recombinant PARP1 was purchased from Biotechne. The His-tagged protein constructs PARP9 macro1 and macro2, PARP15 macro1 and macro2 and LAP3 were expressed and purified as follows: Rosetta 2(DE3) pLysS was used for the PARP9 constructs as well as PARP15macro1 while all remaining constructs were transformed in BL21(DE3). Protein expression was performed at 18 °C for 16 h in presence of 1% ethanol after induction with 0.4 mM IPTG. Bacterial cultures were spun down at $6000 \times g$ for 15 min at 4 °C, followed by resuspension in lysis buffer (30 mM Hepes, pH 7.5; 300 mM NaCl; 5 mM imidazole; 0.2% NP-40; 10% glycerol; proteinase inhibitor cocktail; benzonase; 2 mg lysozyme). Cell suspensions were sonicated on ice using Digital Sonifier 250 Cell Disruptor (Branson). Depending on the expressed protein construct and the pellet size, sonication varied in a range of 2–5 min at 15–20%, with 30 s on and 40 s off. The cell lysates were centrifuged at $12,000 \times g$ for 45 min at 4 °C, to remove cell debris. The supernatant was equipped withTALON Metal Affinity Resin (Takara). Bound proteins were washed multiple times with wash buffer 1 (30 mM Hepes, pH 7.5; 500 mM NaCl; 10 mM imidazole; 0.1% NP-40) followed by wash buffer 2 (30 mM Hepes, pH 7.5; 150 mM NaCl; 10 mM imidazole). After elution (30 mM Hepes, pH 7.5; 300 mM NaCl; 300 mM imidazole; 5% glycerol) proteins were dialysis overnight (30 mM Hepes, pH 7.5; 200 mM NaCl; 5% glycerol) and snap-frozen.

**Protein ADP-ribosylation assays**
Protein ADP-ribosylation assays were routinely carried out at 37°C for 30 min unless indicated otherwise. Reactions were carried out in 30 µl

**Table 4 | Protein purification conditions**

| Protein | Lysis buffer | Wash I | Wash II | Elution buffer | Dialysis buffer |
|---|---|---|---|---|---|
| His-MACROD1 | 20 mM Hepes/NaOH pH 8.0, 300 mM NaCl, 5 mM imidazol, 0.1% NP-40, 10% glycerol, 1 mM TCEP, PIC, benzonase | 20 mM Hepes/NaOH pH 8.0, 300 mM NaCl, 10 mM imidazol, 0.1% NP-40 | | 20 mM Hepes/NaOH pH 8.0, 300 mM NaCl, 300 mM imidazol | 30 mM Hepes/NaOH, pH 8.5, 200 mM NaCl, 10% glycerol, 1 mM TCEP |
| His-MACROD2 | | | | | 20 mM Tris-HCl pH 6.4, 200 mM NaCl, 10% glycerol, 1 mM TCEP |
| His-TARG1 | | | | | 20 mM HEPES/NaOH pH 7.5, 200 mM NaCl, 10% glycerol, 1 mM TCEP |
| His-PARGcat | 20 mM Hepes/NaOH pH 8.0, 300 mM NaCl, 1 mM TCEP | | | 20 mM Hepes/NaOH pH 8.0, 300 mM NaCl, 300 mM imidazol, 1 mM TCEP | 25 mM Hepes/NaOH pH 8.0, 300 mM NaCl, 10% glycerol, 1 mM TCEP |
| His-ARH1 | 50 mM Tris-HCl pH 8.0, 300 mM NaCl, 5 mM MgCl, 1 mM TCEP, 10 mM imidazol, PIC, benzonase | 50 mM Tris-HCl pH 8.0, 150 mM NaCl, 5 mM MgCl, 1 mM TCEP, 10 mM imidazol | 50 mM Tris-HCl pH 8.0, 150 mM NaCl, 5 mM MgCl, 1 mM TCEP, 10 mM imidazol | 50 mM Tris-HCl pH 8.0, 150 mM NaCl, 5 mM MgCl, 1 mM TCEP, 300 mM imidazol | 50 mM Tris/HCl pH 7.0, 150 mM NaCl, 10% glycerol, 1 mM TCEP |
| His-ARH3 | 20 mM Tris-HCl pH 8.0, 150 mM NaCl, 1 mM EDTA, 5 mM DTT, PIC, benzonase | 50 mM Tris-HCl pH 7.0, 150 mM NaCl, 1 mM TCEP, 10 mM imidazol | 50 mM Tris-HCl pH 7.0, 150 mM NaCl, 1 mM TCEP, 10 mM imidazol | 50 mM Tris-HCl pH 7.0, 150 mM NaCl, 1 mM TCEP, 10 mM imidazol | 50 mM Tris/HCl pH 7.0, 150 mM NaCl, 10% glycerol, 1 mM TCEP |
| GST-PARP10_cat | 20 mM Tris-HCl pH 7.5, 300 mM NaCl, 1 mM TCEP, 500 µg/mL lysozyme, PIC, benzonase | 50 mM Tris/HCl pH 8.0, 150 mM NaCl | 50 mM Tris/HCl pH 8.0, 150 mM NaCl | 50 mM Tris/HCl pH 8.0, 150 mM NaCl, 20 mM glutathione | 25 mM Tris/HCl pH 7.5, 150 mM NaCl, 10% glycerol, 1 mM TCEP |
| His-mART2.2 | 25 mM Tris-HCl pH 8.0, 500 mM NaCl, 10% glycerol, 1 mM TCEP, 10 mM imidazol, PIC, benzonase | 50 mM Tris/HCl pH 7.5, 150 mM NaCl, 1 mM TCEP, 10 mM imidazol | 50 mM Tris/HCl pH 7.5, 150 mM NaCl, 1 mM TCEP, 10 mM imidazol | 50 mM Tris/HCl pH 7.5, 150 mM NaCl, 10% glycerol, 250 mM imidazol | 50 mM Tris/HCl pH 7.5, 150 mM NaCl, 10% glycerol, 0.5 mM TCEP |
| His-Pertussis toxin_cat | 25 mM Tris-HCl pH 8.0, 500 mM NaCl, 1 mM TCEP, 10 mM imidazol, PIC, benzonase | 50 mM Tris/HCl pH 8.0, 500 mM NaCl, 1 mM TCEP, 10 mM imidazol | 50 mM Tris/HCl pH 8.0, 500 mM NaCl, 1 mM TCEP, 10 mM imidazol | 25 mM Tris/HCl pH 8.0, 500 mM NaCl, 300 mM imidazol | 25 mM Hepes/NaOH, pH 8.0, 500 mM NaCl, 10% glycerol, 1 mM TCEP |

containing 50 mM Tris–HCl, pH 7.5, 0.5 mM DTT, 0.1% Triton X-100, 5 mM MgCl2, and 50 or 500 µM β-NAD$^+$ (Sigma-Aldrich). 2 µl HeLa cytosolic extract was used for the mART2.2 reactions. The amounts of transferases used varied depending on the activity of the enzyme studied as indicated in the figures. Reactions were placed on ice before adding hydrolase proteins, and were indicated PARP inhibitors were added. Hydrolase reactions were incubated at 37°C for 30 min unless indicated otherwise, stopped by adding SDS sample buffer and separated using SDS–PAGE. Samples were analysed using precast 4-20% gradient gels (BioRad) or homemade 15% tris-glycine gels. ADP-ribosylation levels were determined following western blotting of the proteins onto nitrocellulose using RTA nitrocellulose transfer stacks (Biorad). An anti-PAR/MAR antibody (E6F6A, Cell Signaling Technology) was used at a 1:100.000 dilution. Western blots were incubated with WesternBright Quantum HRP substrate (Advansta) and chemiluminescence was detected using the Azure600.

**RNA/DNA ADP-ribosylation and hydrolase assays**

For RNA modification, 10 µM of ssRNA (5'P-GUUUCGGAUCGACGC-Cy3) was incubated with 10 µM TRPT1 or PARP10cat in ADPr-reaction buffer (20 mM Hepes–KOH, pH 7.6, 50 mM KCl, 5 mM MgCl2, 1U/µL RNase inhibitor) and 500 µM NAD at 37 °C for 90 min while shaking. TRPT1 was digested by proteinase K treatment (20 U), for 20 min at RT, before purification using the Monarch RNA Cleanup Kit (T2030). Concentration was determined by spectrophotometric measurements (NanoDrop ND-1000 Spectrophotometer). For standard RNA hydrolase assays the indicated amounts of purified substrate oligos were incubated with the indicated amounts of hydrolases in ADPr-buffer at 37 °C for 30–40 min. After the reaction time, proteins were digested with 20 U Proteinase K (New England Biolabs) for 10 min at RT. ADP-ribosyl or hydrolase assay samples were completed with RNA loading dye (New England Biolabs), heated up for 3 min at 85 °C and loaded on pre-run urea polyacrylamide gels (8 M urea, 10–15% acrylamide:bisacrylamide (19:1), 0.2% APS, 0.4% TEMED). Gels were run in TBE buffer (89 mM Tris, pH 8.0, 89 mM boric acid, 2 mM EDTA) at 7 W. The oligos were visualized using the Azure 600 equipment either through Cy3-labels or by staining with SYBR™ Gold nucleic acid gel stain (Invitrogen).

For the comparison of ADPr-RNA de-capping activity of different hydrolase time course experiment was performed where de-capping reaction was monitored by visalisation using PAGE. RNA-3′-Cy3 labelled was first ADP-ribosylated using TRPT1 and then purified aftere proteinase K digestion. De-capping assay was then preformed at 1 uM ADPr-RNA substrate concentration and indicated enzyme concentrations. Apperaence of de-capped RNA-3′-Cy3 was monitored over time using in gel fluorescence measurements and quantified using ImageJ[61]. Due to the fixed and saturating substrate concentration used in our assays, the traditional approach of varying substrate concentration to obtain Michaelis-Menten kinetics was not employed. Instead, the following kinetic parameters were estimated from the time-course data. The initial reaction velocity ($V_0$) was estimated from the linear portion of the time-course data, where the product formation rate is constant, and substrate depletion is negligible. Given that the substrate concentration was saturating the apparent rate constant ($k_{app}$) was calculated using the initial velocity divided by the enzyme concentration:

$$k_{app} = \frac{V_0}{(E)}$$

where $V_0$ is the initial reaction velocity, and [E] is the enzyme concentration.

Maximum Reaction Velocity ($V_{max}$) was approximated by the plateau of the product formation curve and then turnover number ($k_{cat}$) was derived by normalizing $V_{max}$ to the enzyme concentration, giving the number of substrate molecules converted to product per second per enzyme molecule:

$$k_{cat} = \frac{V_{max}}{(E)}$$

This value is a reflection of the catalytic activity of the enzyme under the assay conditions and enable us direct comparison of the hydrolase activities. Data analysis was performed using R statistical package[62].

## Cell lines and transfections

HEK293, HeLa, Ln229 and U2OS cells (ATCC) were cultured in dulbecco's modified eagle medium (DMEM) with 4.5 g/L glucose, pyruvate and 10% heat-inactivated FBS (Gibco), RD cells were cultured under the same conditions but in a 1 g/L glucose medium. Cells were kept at 37 °C in a humidified atmosphere with 5% $CO_2$. siRNA transfections were performed using RNAiMax (Invitrogen) according to the manufacturer's instructions. Cells were washed 5–6 h after transfection, and analysed 48–72 h later. siRNA SMARTpools targeting TARG1, PARG or ARH3 were purchased from Dharmacon. A construct encoding for mEGFP-tagged full-length PARG was transfected into HEK293 cells using calcium phosphate.

## Cell lysis, GFP-IP and western blot

Cell protein extractions were performed using RIPA buffer (150 mM NaCl, 1% Triton X-100, 0.5% sodium deoxycholate, 0.1% SDS, 50 mM Tris-HCl (pH 8.0)) supplemented with protease inhibitor cocktail (Sigma) and olaparib (Selleck Chemicals), followed by sonication on ice or benzonase treatment (Sigma). Lysates were centrifuged at 13,000 xg and supernatants collected. After addition of SDS-sample buffer, the samples were run on tris-glycine SDS-polyacrylamide gels and subsequently blotted onto nitrocellulose membranes. Membranes were blocked with 5% non-fat milk in TBST for 1 hr at RT, primary antibodies were diluted in TBST and incubated overnight at 4 °C, secondary antibodies were diluted in 5% non-fat milk in TBST and incubated for 1 h at RT. Wash steps were performed in between and after antibody incubations with TBST at RT for at least 5 mins. Chemiluminescent signals generated upon addition of WesternBright Quantum HRP substrate (Advansta) were detected using the Azure600. Antibodies used are PARG (1:5000, Cell Signaling Technology), MAC-ROD2 (1:100, homemade), PAR/MAR (1:50,000 Cell Signaling Technology) and tubulin (1:1000, Santa Cruz). To enrich PARG from HEK293T cells, the cells were lysed in CoIP buffer (10 mM HEPES pH 7.5, 50 mM NaCl, 0,20% Triton X-100, 10% glycerol, protease inhibitor cocktail and olaparib), followed by centrifugation for 10 min at $13,000 \times g$. The supernatant was incubated with 15 µL GFP-magnetic agarose (Chromotek) per 10 cm dish at 4 °C for 1 h. Following the IP, beads were washed extensively before using the immunoprecipitated protein in enzymatic assays.

## Cellular RNA Slot blot

RNA was blotted on Hybond-N membrane (Amersham) using a PR648 Slot Blot Blotting Manifold (Hoefer). The membrane was pre-wetted in SSC buffer, pH 7.0 (150 mM NaCl and 15 mM NaCit), for 5 min. After assembling of the blotting sandwich, slots were flushed with SSC buffer. 400 ng total RNA in 200 µL SSC buffer per slot was applied to the membrane. Samples were cross-linked using UV light (120 mJ/cm²) before the membrane was air-dried, and samples were cross-linked once more using UV light (120 mJ/cm²). The membrane was blocked with 5% non-fat milk in PBST for 60 min at RT, anti-PAR/MAR antibody (E6F6A, Cell Signaling Technology) was used at a 1:50,000 dilution diluted in PBST and incubated overnight at 4 °C, and secondary antibodies were diluted 1:10,000 in 2% non-fat milk in PBST and incubated for 30 min at RT. Multiple wash steps were performed both in between and after the antibody incubation with PBST for at least 5 min. Chemiluminescent signals were detected using the Azure 600 equipment.

## Statistics and reproducibility

All experiments relying on purified proteins and biochemical readouts were replicated at least once, whereas all experiments using cultured cells were performed at least three times to ensure reproducibility. The base R package and ggplot2 were used for processing and plotting of data[62,63].

## Reporting summary

Further information on research design is available in the Nature Portfolio Reporting Summary linked to this article.

## Data availability

All relevant data are available with this publication and its supplementary files, with uncropped blots and gels available in the Supplementary Data file 1 and numerical data for AMP Glo measurements in Supplementary Data File 2. The plasmids generated here are available from Addgene: pDONR-PARG 219956, pCDNA5/FRT/TO-N-mEGFP-PARG 217663, pDONR-NUDT5 219954, pDEST17-NUDT5 219955, pDONR-LAP3 217660, pDEST17-LAP3 217661.

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

## Acknowledgements
We thank all the members of our lab as well as Michael Hottiger and Michael Cohen for fruitful discussions and Patricia Korn, Lari Lehtiö, and Fritz Koch-Nolte for sharing plasmids. Recombinant PARP14macro1, -macro2 and -macro3 were a gift from Ribon Therapeutics, recombinant Giα was a gift from Ivan Rodriguez Siordia and Michael Cohen. This work was supported by the Proteomics Facility, a core facility of the Interdisciplinary Centre for Clinical Research (IZKF) Aachen within the Faculty of Medicine at RWTH Aachen University. This work was funded by the Deutsche Forschungsgemeinschaft to K.F.Ž. (FE1423/3-1), by the START program of the Faculty of Medicine, RWTH Aachen University to R.Ž. (13/20), to L.W. (121/22) and to K.F.Ž. (116/22) and by the Habilitation Stipendium of the Faculty of Medicine, RWTH Aachen University to K.F.Ž.

## Author contributions
Investigation: L.W., R.Ž., N.I., J.S., G.M., and K.F.Ž. Resources: G.A., S.W., D.F., and B.L. Conceptualisation, formal analysis, supervision and visualisation: R.Ž. and K.F.Ž. Writing – original draft: K.F.Ž. Writing – review & editing: all authors. All authors agreed with the final manuscript.

## Funding

## Competing interests
The authors declare no competing interests.
