## [Transparent Peer Review file · Communications Biology]

Family-wide analysis of human macrodomains reveals novel activities and identifies PARG as most efficient ADPr-RNA hydrolase

Corresponding Author: Dr Karla Feijs-Žaja

This manuscript has been previously reviewed at another journal. This document only contains information relating to versions considered at Communications Biology.

Version 0:

Reviewer comments:

Reviewer #1

(Remarks to the Author)

The manuscript by Weixler describes the comprehensive functional analysis of proteins containing macrodomains. In particular, the authors examined the ADP-ribose (ADPr) hydrolase activity of macrodomains against various protein and peptide substrates with defined amino acid ADPr-linkages. This analysis not only confirmed the recently described Glu/Asp-specific ADPr activity of PARP9macro1 and PARP14macro1, but also revealed that PARP14macro2 and PARP15macro2 are active ADPr hydrolases and not just ADPr readers as previously thought. The remove the ADPr group on auto-modified pertussis toxin; however, the identity of the amino acid-ADPr linkage is unknown. For the first time, the authors also show that PARP9macro1 and PARP14macro1 can act as ADPr-RNA (attached the the 5'-phosphate) decapping enzymes. RNA ADP-ribosylation is a relatively new and exciting type of post-translational modification of RNA that likely impacts various aspects of RNA function. Finally, the authors provide compelling evidence, both in vitro and in cells, that PARG is an efficient ADPr-RNA decapping enzyme and is the major ADPr-RNA decapping enzyme in cells.

Overall, the manuscript is well-written, and the experiments are well-executed. The rising interest in ADP-ribosylation and the critical role of macrodomains in regulating ADP-ribosylation dynamics will make this manuscript of interest to the broad readership of Communications Biology. I recommend that the following minor issues be addressed before publication.

Specific points:

1. The authors should replace "RNA decapping" with "ADPr-RNA decapping."
2. Have the authors tested if hydroxylamine (pH9) or mild basic conditions remove the ADPr from auto-modified pertussis toxin? This could indicate lysine-ADPr.
3. While likely beyond the scope of the paper have the authors considered using MS/MS to identify the auto-modification site in pertussis toxin?
4. Do other macrodomain enzymes play a more prominent role in regulating ADPr-RNA levels in cells that have lower PARG levels, such as U2OS cells?

Reviewer #2

(Remarks to the Author)

Weixler et al. has screened a series of human macrodomains for their ADP-ribosylation hydrolase activities and RNA decapping activities. They reported that PARP9macro1 and PARP14macro1 can reverse ADP-ribosylation from acidic residues, and PARP14macro2 and PARP15macro2 can function as ADP-ribosylhydrolases. In addition, they identified both PARP9macro1 and PARP14macro1 as RNA decapping protein domains.

Strengths: Overall, this study is very interesting as it explores the ADP-ribosylation hydrolase activities and RNA decapping activities roles of different macrodomain containing proteins.

Weakness: The current study initially used FoldSeek tool screening approach to identify LAP3 as a potential macrodomain-like fold containing protein, and then identified a few known MDCPs based on the structural alignment with LAP3. However, LAP3 doesn't show either hydrolase activities or RNA decapping activities towards any of the substrates by in vitro assays. Can the authors provide some explanations or discussions about this approach?

Many of the macrodomains tested in Fig.2 seem able to hydrolyze glutamate-ADPr on PARP10, but only some of them towards arginine, cysteine or other amino acids. Can the authors provide more evidence to test the amino acid specificity, possibly by using synthesized ADPr-peptide?

It is better to test the hydrolase activities and RNA decapping activities of the macrodomain containing proteins they screened in cell-based assay and explore their biological significance.

Reviewer #3

(Remarks to the Author)

In this paper, Weixler et al. systematically describes the human macrodomains and their activities. They find a putative novel macrodomain-containing protein, LAP3, and furthermore expands on catalytic activity for other of the macrodomains. Overall, the majority of the paper is well-written, and easy to follow. The section 'PARG is the most effective RNA ADP-ribosylhydrolase in vitro' is a bit difficult to read. I appreciate the authors' honesty and self-awareness regarding limitations. The story lack some novelty, and would benefit from adding a few follow-up experiments – but overall a fine story.

General points:

- The authors dedicate the first figure to the putative macrodomain-containing protein LAP3. It would be helpful to include a few sentences, either in the introduction, results, or discussion, about what is known about this protein. Finding a hydrolase function for LAP3 would be exciting. I understand that this is not trivial, but to my understanding, it was only included and tested in the experiments in relation to Figure 2.
- An essential element when testing hydrolase activity and specificity is substrates. The authors use glutamate-, arginine-, and cysteine-linked ADP-ribosylation together with poly(ADP-ribose). Having more linkages available, and potentially also several different substrates for the different linkages would add important knowledge. Several labs are able to produce (semi)synthetic ADP-ribosylated peptides, that would be a useful tool for studying this. Additionally, the reaction with PARP1 should be performed +/- HPF1.
- Personally, I like having loading controls for the western blots in the main figures.

Comments to specific pages:

- Page 1: the authors mention that PARP7 has been reported to mainly modify cysteine residues (and to a lesser extent arginine and tyrosine). However, the authors could mention the paper from Parsons et al., 2021, where they mainly identify glutamic and aspartic acid residues as being modified.
- Page 2: the authors discuss that reversal of ADP-ribosylation is essential for cell viability, and gives numerous examples of this. They could also mention that PARG knockout has been shown to be embryonic lethal (Koh et al., 2004).
- Page 3, Figure 1c: Model confidence is unreadable, please change the legend.
- Page 3: the authors use recombinant PARP1 to produce poly(ADP-ribose). As also mentioned above, the experiment with PARP1 + HPF1 is missing. Maybe it would even be possible to first incubate with PARG to generate mono(ADP-ribose), and then with other hydrolases after?
- Page 4: The sentence 'In accordance with previous studies only PARG and ARH3 exhibit glycohydrolytic activity toward a PARylated substrate.' needs a reference. ARH3 is normally thought to remove serine-linked ADP-ribosylation, and has the highest activity towards MAR. Doesn't Figure 3f suggest this as well?
- Page 6, Figure 3b: Please confirm that Gi-alpha is ADP-ribosylated on cysteine residue for example by treatment with mercury chloride, which should remove the modification.
- Page 8: This section is a bit difficult to read. Maybe it could be beneficial to collect all the numbers in a table instead.
- Page 11, Figure 7c: the authors include RD cells. Please indicate in the text why this is a good control.
- Page 11: the authors write 'The hydrolases may have different activities depending on the exact substrate...' It would therefore be interesting to include several substrates modified on each amino acid residue.
- Page 12: the authors are missing an 'of'; '...classic hydrolysis through formation of an oxocarbenium intermediate...'

Version 1:

Reviewer comments:

Reviewer #1

(Remarks to the Author)

The authors have sufficiently addressed my comments.

Reviewer #3

(Remarks to the Author)
